# Proteomic Identification and Characterization of Collagen from Bactrian Camel (*Camelus bactrianus*) Hoof

**DOI:** 10.3390/foods12173303

**Published:** 2023-09-02

**Authors:** Yingli Wang, Le Song, Chengcheng Guo, Rimutu Ji

**Affiliations:** 1College of Food Science and Engineering, Inner Mongolia Agricultural University, Hohhot 010018, China; wyli2008@163.com (Y.W.); imau15849197417@126.com (L.S.); mocc2018@126.com (C.G.); 2Inner Mongolia Institute of Camel Research, Alxa 737300, China

**Keywords:** Bactrian camel hoof, collagen, proteomic analysis, characterization

## Abstract

With the development of camel-derived food and pharmaceutical cosmetics, camel hoof, as a unique by-product of the camel industry, has gradually attracted the attention of scientific researchers in the fields of nutrition, health care, and biomaterial development. In this study, the protein composition and collagen type of Bactrian camel hoof collagen extract (CHC) were analyzed by LC-MS/MS, and the functional properties of CHC were further investigated, including its rheological characteristics, emulsification and emulsion stability, and hygroscopicity and humectancy. Proteomic identification confirmed that CHC had 13 collagen subunits, dominated by type I collagen (α1, α2), with molecular weights mainly in the 100–200 KDa range and a pI of 7.48. An amino acid study of CHC revealed that it carried the standard amino acid profile of type I collagen and was abundant in Gly, Pro, Glu, Ala, and Arg. Additionally, studies using circular dichroism spectroscopy and Fourier transform infrared spectroscopy revealed that CHC contains a collagen-like triple helix structure that is stable and intact. Different concentrations of CHC solutions showed shear-thinning flow behavior. Its tan δ did not differ much with increasing concentration. The CHC has good emulsifying ability and stability, humectancy, and hygroscopicity. This study provides a basis for utilizing and developing Bactrian camel hoof collagen as a functional ingredient.

## 1. Introduction

Collagen is one of the most prevalent animal proteins, with substantial quantities found in the skin, bone, cartilage, and tendon, accounting for around 30% of total animal protein [1,2]. Collagen is the central extracellular matrix molecule that affects the mechanical flexibility of connective tissues and is believed to be the biological glue that binds cells in place [3]. Modern medical research has confirmed that collagen induces platelet adhesion and aggregation, releases coagulation factors, and promotes blood coagulation. It is one of the most commonly used materials in protein scaffolds for skin wound healing [4,5]. Due to its superior biological qualities, including low immunogenicity, compatibility, degradability, coagulability, and proliferation ability, collagen is also frequently utilized in biomedical materials, tissue engineering, burns, cosmetology, food, and other industries [5,6,7]. In addition, due to the high molecular weight of collagen, different types of collagen have different compositions, forming a functionally diverse collagen family. Researchers have extracted collagen from various animal bodies, including pig skin, bovine Achilles tendon, sheep skin, fish skin, scales, etc. [8,9].

Bactrian camel (*Camelus bactrianus*) hooves are a unique and superior food from the Alashan region of China. Camel hoof is sweet, salty, flat, and more precious than camel meat. Its meat is delicate, elastic, and soft [10]. It has developed its unique physiological function of resisting extreme environments during long-term natural selection. The hoof is the most active part of the camel body, consisting of large, soft finger pillows at the rear end and small, pointed toes at the front end. The content of Ca, Na, Fe, Zn, and other elements in camel hoof is significantly higher than that of eggs [11]. Compared with pig hooves and cow hooves, camel hooves are characterized by high protein content and low oil content, in which the contents of some nutrients contained are 32.8 g of protein, 2 g of fat, 26 μg of vitamin A, 170.6 mg of sodium, 152 mg of calcium, 59 mg of magnesium, and 261 mg of copper per 100 g [11]. The medical benefits of camel hoof may include reducing fatigue, improving skin, preserving youthful beauty, etc. [10,11]. However, there are fewer reports of related research in this area. Scientific research has proven in recent years that there is very little connection between disorders, including mad cow disease, and collagen of animal origin [2]. Less comprehensive information is available on the purification and characterization of collagen from camel hooves. Due to widespread slaughter in pastoral areas, the annual stock of camel hooves is relatively considerable. Still, China has long lacked this “best” raw material for processing and producing camels. Twenty-eight types of collagen have been discovered, and their structural characterization, functional properties [12], and medicinal functions are still in the exploratory and discovery stage due to their widespread presence in living organisms. Bactrian camel hoof collagen, as a result, can be used as a new research direction for collagen bio-matrix raw materials.

In this study, CHC was obtained from camel hoof using enzyme-acid hydrolysis, the main types and amounts of collagen were analyzed by protein technology combined with liquid chromatography-tandem mass spectrometry, and further analyses of the structural characterization and functional properties of CHC were carried out. The findings of this research will offer an innovative strategy for the high-value use and development of camel hoof proteins as well as a theoretical foundation for the application and advancement of CHC and collagen peptides as functional components.

## 2. Materials and Methods

### 2.1. Materials and Chemicals

Camel hooves were obtained from six Bactrian camels (9 to 10 years old) available at local abattoirs (Alashan, Inner Mongolia, China). Camel hooves were depilated and peeled, bone was removed, and they were vacuum-packed in polyvinylidene chloride bags and stored at −25 °C for further chemical composition. Pepsin (250 N.F.U/mg) and hydroxyproline (HYP) standards were purchased from Beijing Solaria Technology Co., Ltd., Beijing, China. Other reagents were of analytical grade and obtained from Sinopharm Chemical Reagent Co., Ltd., Shanghai, China.

### 2.2. Sample Preparation of CHC

Camel hoof samples were cut into small pieces and shaken in ice water with 10% 1-butanol (1:15 *w*/*v*) for 24 h to remove fat in fluid changed every 6 h. The residue was then collected by centrifugation at 8000× *g* for 10 min at 4 °C and repeatedly washed with ice water to wipe off the fat. The leftover material was soaked in 0.1 M NaOH (1:10 *w*/*v*) for 24 h, with fluid changes every 6 h, to eliminate non-collagenous proteins and pigments. The residue was then washed with deionized water and centrifuged at 8000× *g* for 10 min. Finally, the pretreated Bactrian camel hoof sample was placed at −20 °C until use. Skimmed Bactrian camel hoof samples were mixed with 0.5 mol/L glacial acetic acid at a ratio of 1:30 (*w*/*v*). The substrate was hydrolyzed by pepsin (4000 u/g powder) for 48 h at 4 °C. The supernatant was centrifuged at 12,000× *g* for 20 min, and sodium chloride was added to the supernatant to a concentration of 0.9 mol/L. The supernatant was stirred at a low temperature for 24 h. The supernatant was centrifuged at 10,000× *g* for 20 min to isolate the collagen. After being entirely dissolved in 0.5 mol/L acetic acid solution, dialysis was carried out for 24 h. The sample solution was frozen. An HYP kit was used to detect the HYP content, and the collagen yield [13] was estimated using the following formula:(1)$$\mathrm{Yield}(\%)=\frac{\mathrm{Amount}\;\mathrm{of}\;\mathrm{Hyp}\;\mathrm{in}\;\mathrm{collagen}\;(\mathrm{mg})}{\mathrm{Amount}\;\mathrm{of}\;\mathrm{Hyp}\;\mathrm{in}\;\mathrm{Bactrian}\;\mathrm{camel}\;\mathrm{hoof}\;(\mathrm{mg})}\times 100\%$$

### 2.3. Amino Acid Analysis

Amino acid content was assessed based on the method of Song et al. [10] was slightly modified. CHC samples were completely hydrolyzed to free amino acids by 6 mol/L HCl, proportionally diluted and evaporated to dryness by nitrogen blowing, solubilized in 0.02 mol/L HCl, filtered through a 0.22 μm pore size, and then analyzed using a high-speed analyzer (L-8900; Hitachi, Tokyo, Japan) for 20 µL of the filtrate.

### 2.4. Camel Hoof Protein Analysis

The protein enzymatic hydrolysis process of camel hoof was carried out, as previously reported by Song et al. [10]. Briefly, the tissue samples were first ground with liquid nitrogen, followed by ultrasonic treatment with 300 μL SDT buffer solution (4% SDS,100 mM Tris-HCl) lysate in a boiling water bath and centrifuged at 10,000× *g* for 10 min to collect supernatant, and then an appropriate amount of 1M DL-dithiothreitol (DTT) was added for proteolytic hydrolysis using filter-aided sample preparation (FASP) method. A 200 μL urea (UA) buffer solution was added to the protein enzymatic hydrolysate and centrifuged at 12,000× *g* for 15 min twice. An appropriate amount of iodoacetamide (IAA) was added to the residue, reacted without light for 30 min, and centrifuged at 12,000× *g* for 15 min. A 100 μL UA buffer solution was added and centrifuged at 12,000× *g* for 15 min twice. A 100 μL ammonium bicarbonate buffer solution was added and centrifuged at 14,000× *g* for 15 min twice. A 40 μL trypsin buffer solution was added, reacted at 37 °C for 16–18 h, and centrifuged at 12,000× *g* for 15 min. The filtrate was collected, desalted with C18 Stage Tip, and redissolved with 0.1% trifluoroacetic acid (TFA) after vacuum drying to obtain the peptide samples.

Peptide samples were separated using an Easy nLC 1200 Chromatography System (Thermo Fisher Scientific Co. (Shanghai, China)) at a nanolitre flow rate. The column was balanced with 100% liquid A (0.1% formic acid aqueous solution). After the samples were injected into the Trap Column, the chromatographic column was used for gradient separation with a flow rate of 300 nL/min. Liquid B is a mixture of 0.1% formic acid, acetonitrile (80%), and water, and the gradient of liquid B is as follows: 1% (0–3 min), 2–8% (3–43 min), 8–28% (43–51 min), 28–40% (51–52 min), 40–100% (52–60 min), and maintained at 100%. Data-dependent acquisition mass spectrometry was carried out using a Q-Exactive HF-X mass spectrometer (Thermo Fisher Scientific Co. (Shanghai, China)). The detection mode is a positive ion, the scanning range of the parent ion is 350–1800 m/z, the primary mass spectrometry resolution is 60,000 @m/z 200, and the automatic gain control (AGC) target is 3·e6. The direct Maximum IT is 50 ms. Secondary mass spectrometry of peptide segment triggers the collection of secondary mass spectra of 20 highest-intensity parent ions after each full scan. Secondary mass spectrometry resolution 15,000 @m/z 200, the AGC target is 1·e5, the second-level Maximum IT is 25 ms, the MS2 ActivationType is high-energy collision dissociation (HCD), and the Isolation window is 1.6 m/z. Normalized collision energy is 28. MaxQuant1.6.1.0; a mass spectrometry database retrieval tool was used for the analysis. Camelus bactrianus (Camel) served as the reference protein database, containing 21,155 protein sequences. The download address is https://www.uniprot.org/taxonomy/9837 (accessed on 27 August 2020). Highly reliable identification results were obtained by screening false discovery rate (FDR) < 0.01 of protein and FDR < 0.01 of peptide-spectrum matches (PSM).

### 2.5. Characterization of Collagen Extraction

#### 2.5.1. Fourier Transform Infrared Spectroscopy

The method of Zou et al. [13] was slightly modified. The CHC was mixed with KBr and extruded onto a transparent sheet. The 4000–400 cm^−1^ FTIR spectra were recorded with an FTIR spectrometer (Thermo Fisher Scientific Co. (Shanghai, China)).

#### 2.5.2. Circular Dichroism Spectra

As Li et al. [11] described, the CHC was prepared in 0.5 mol/L glacial acetic acid at a concentration of 0.3 mg/mL. The sample was detected at intervals of 2 nm in the 190–250 nm using the Jasco J-815 system (Japan Spectroscopic Company, Tokyo, Japan).

### 2.6. Analysis of Collagen Properties

#### 2.6.1. Emulsification and Emulsion Stability

We took 10 mL CHC solution of different concentrations with pH adjusted to 7, mixed it with 10 mL peanut oil, centrifuged at 1000× *g* for 5 min, and recorded the emulsion layer’s volume (or height). Meanwhile, 20 mL of the above sample emulsion was placed in a 50 °C water bath, and the height of the emulsion layer was recorded after 30 min. The following formulas were used to determine the emulsification and emulsion stability. Additionally, emulsification and emulsion stability of CHC solution (3 mol/mL) were examined at pH 3–11 and different concentrations of NaC1, respectively, according to the above-described method.
(2)$$\mathrm{Emulsification}(\%)=\frac{\mathrm{Volume}\;\mathrm{of}\;\mathrm{emulsion}\;\mathrm{layer}}{\mathrm{Initial}\;\mathrm{volume}\;\mathrm{of}\;\mathrm{sample}\;\mathrm{liquid}}\times 100\%$$
(3)$$\mathrm{Emulsion}\;\mathrm{stability}\;(\%)\;=\frac{\mathrm{Height}\;\mathrm{of}\;\mathrm{liquid}\;\mathrm{in}\;\mathrm{emulsified}\;\mathrm{state}\;}{\;\mathrm{Initial}\;\mathrm{height}\;\mathrm{of}\;\mathrm{emulsion}}\times 100\%$$

#### 2.6.2. Hygroscopicity and Humectancy

A 0.200 g collagen sample was placed in a constant weight weighing bottle in a sealed continuous temperature and humidity incubator (30 °C, 80% humidity), and the mass of the sample was measured accurately at regular intervals. At the same time, 0.200 g of collagen sample was placed in a weighing flask with 10% distilled water in a sealed desiccator containing anhydrous silica gel, and the weight of the sample was measured at regular intervals. The above experimental procedure was carried out using glycerol as a control and the following formulas: moisture absorption and retention were calculated as follows:(4)$$\mathrm{Hygroscopicity}\;(\%)\;=\frac{\;\left(\mathrm{Weight}\;\mathrm{of}\;\mathrm{sample}\;t-\;\mathrm{hour}-\;\mathrm{Initial}\;\mathrm{weight}\;\mathrm{of}\;\mathrm{sample}\right)\;}{\mathrm{Initial}\;\mathrm{weight}\;\mathrm{of}\;\mathrm{sample}}\times 100\%$$
(5)$$\mathrm{Humectancy}\;(\%)\;=\frac{\;\left(\mathrm{Initial}\;\mathrm{moisture}\;\mathrm{weight}\;\mathrm{of}\;\mathrm{sample}\;-\;\mathrm{moisture}\;\mathrm{weight}\;\mathrm{of}\;\mathrm{sample}\;t-\mathrm{hour}\right)\;}{\mathrm{Initial}\;\mathrm{moisture}\;\mathrm{weight}\;\mathrm{of}\;\mathrm{sample}}\times 100\%$$

#### 2.6.3. Rheological Properties

The rheological properties were determined using the DHR-2 rotational rheometer cone-plate measurement system (TA Instruments Inc., New Castle, DE, USA). These measurements were obtained as described in a previous study [14]. The experimental parameters were a scanning frequency of 0.1–80 Hz and a parallel stainless steel plate abrasive with a diameter of 40 mm and a 1° clamping angle.

### 2.7. Statistical Analysis

The findings in this study are presented as mean ± standard deviation (± SD), with all measures being performed in triplicate. Data were analyzed using one-way ANOVA in SPSS 25.0 (SPSS Inc., Chicago, IL, USA) to evaluate whether there were significant differences between the samples. A *p* value of < 0.05 was considered significant. Protein theoretical isoelectric point (pI) and molecular weight (Mw) were obtained using Compute pI/Mw tool. The analysis used Microsoft Excel 2019 (Microsoft Co., Redmond, WA, USA) software and GraphPad Prism 9.5 (GraphPad Software Inc., La Jolla, CA, USA) to generate graphs.

## 3. Results and Discussion

### 3.1. Yield of CHC

The appropriate concentration of glacial acetic acid solution facilitates the extraction of CHC. This may be because the low-acidic environment interfered with the stability of the salt bonds and Schiff base structure between collagen molecules, facilitating collagen solubilization [15]. In this experiment, the collagen yield of CHC was 35.77 ± 2.06%. This is similar to the results of collagen extraction from Bactrian camel skin [16].

### 3.2. Proteome Identification

After applying the protein identification screening criteria of FDR ≤ 0.01, 121 proteins were successfully identified from the CHC samples. Among these proteins, there were 13 collagen subunits and precollagen-enhancing factors (Table 1). The top three identified proteins in CHC were ANK, collagen alpha-1(I) chain, and collagen alpha-2(I) chain. ANKs are protein modules with β2α2 structures widely found in all organisms, and they can bind to various ligands to realize complex biological functions, such as protein modification, protein ubiquitination, etc. [17,18]. ANK can rely on hydrogen bonds and hydrophobic interactions to form a tightly stabilized structure with collagen macromolecules, as shown by the 32.60% iBAQ percentage of ANKs in CHC [18].

Eighty collagen peptides were identified in CHC, and the identified collagens were collagen type Ⅺ (α1, α2), collagen type I (α1, α2), collagen type II (α1), collagen type III (α1), collagen type IV (α2), collagen type V (α1, α2, α3), collagen type VI (α1, α2, α3), and the precollagen c endopeptidase enhancer (Appendix A). Most of these collagens are associated with calcium binding, metal linking, extracellular matrix organization, and the assembly of collagen fibrils and other multimers [2]. Type I collagen is mainly involved in forming protofibrils in tendons, ligaments, and bones [19]. Type III collagen and type VI collagen are primarily associated with developing type III and V collagen trimers and assembling collagen fibers and other multimeric structures [20,21]. The CHC had the highest percentage (Figure 1) of type I collagen (48.85% for IBAQ), followed by type III collagen (α1) and type VI collagen (α1, α2, α3). In earlier investigations, these experimental findings were not covered. Variations in the type and amount of collagen in CHC may result from a variety of factors, such as the replacement of proteins during camel growth, carrying a large body, and environmental conditions [2,22].

Figure 1 summarizes the molecular weight, pI, and iBAQ % distributions of CHC and identified collagens. The structural characteristics and processing applications of collagen are directly related to its isoelectric point (pI). The pI of the identified collagen α1 chains were 9.28 (type I), 5.96 (type III), 8.38 (type II), 5.18 (type VI), 4.84 (type V), 5.66 (type XI fragment), with a mean of 6.55. The pI of the collagen α2 chains were 9.30 (type I), 9.36 (type IV), 5.54 (type VI), 7.95 (type V), and 5.82 (type XI), with a mean of 7.59. The pI of collagen α3 chains were 5.87 (type VI) and 5.82 (type V), and the mean value was 5.85. The mean pI value of camel hoof collagen type I was 9.29, significantly higher than that of bovine pericardium type I bovine collagen and bovine pure type I triple helical chain [23]. Compared to bovine trotters and bone collagen, the mean PI of collagen in CHC was 6.84 [19,20]. Due to the positive or negative charge it gives the sample, a pH value lower than 4.0 or higher than 10.0 is acceptable for the purification process of CHC.

Minor collagen structure variations can be observed in tissues derived from the same animal or across various animal species. Due to these potential variations, it is possible to separate collagen molecules of different weights [23]. The collagen molecular weights in CHC were mainly distributed in the range of 100–200 kDa (iBAQ of 49.65%), with the most prominent being collagen type VI α3 (339.58 kDa) and the smallest being collagen type XI α1 fragment (89.147 kDa). The molecular weights of type I collagen in CHC were 138.94 kDa (α1) and 120.28 kDa (α2), which were slightly higher than the molecular weights of the α1 and α2 chains of pure bovine type I collagen [1]. This also indicates that the collagen of the camel hoof was not extensively degraded or dissolved during the extraction process. In this study, collagen α1 chains in CHC were mainly from type I, type II (141.83 kDa), type III (138.44 kDa), type V (93.742 kDa), type VI (108.67 kDa), and type XI fragment (89.147 kDa), with an average molecular weight of 114.47 kDa, and the α2 chains were mainly from type I, type IV (147.58 kDa), type V (123.7 kDa), type VI (97.135 kDa), and type XI (172.24 kDa), with an average molecular weight of 132.19 kDa. There are also two α3 chains, from type VI and type V (172.07), with an average molecular weight of 255.83 kDa, and this may be the trimeric structure of the collagen α-chain, which is often referred to as the γ-chain [10].

This further proves that the camel hoof collagen molecule consists of the α1 chain, α2 chain, and α3 chain. The strength of the α1 chain of CHC (26.39%) was higher than that of the α2 chain (22.89%), while the α3 chain was only 0.09%. This is consistent with previous polyacrylamide gel studies. Additionally found was procollagen C-endopeptidase enhancer (48.21 kDa). These findings imply that the structural stability of camel hoof tissue is considerably maintained by the 1 and 2 chains of collagen, particularly from collagen I and II.

### 3.3. CHC Amino Acid Composition

The amino acid composition of CHC is listed in Table 2. CHC had the highest glycine (15.11 ± 1.78 g/100 g), followed by proline (9.54 ± 1.06 g/100 g), glutamic acid (7.28 ± 0.46 g/100 g), alanine (6.20 ± 0.38 g/100 g), and hydroxyproline (6.06 ± 1.16 g/100 g). The total content of hydroxyproline and proline was found to be 15.61 ± 1.28 g/100 g. In contrast to most proteins, collagen contains nearly 1/3 of glycine and is free of cysteine and tryptophan [24]. The sample comprised 153 proline, 97 hydroxyproline, and 243 glycine per 1000 residues of amino acids. Its tyrosine, histidine, and methionine contents were minimal, and neither cysteine nor tryptophan could be found. The observed result aligns with the normal arrangement of collagen in mammals [2]. In addition, the high content of polar amino acids is also a characteristic of collagen amino acids [6,23,25]. The polar amino acid content of CHC was 19.53 ± 1.15 g/100 g, representing 31.39% of the total amino acid content. This finding is significantly higher than fish skin and bovine Achilles tendon [19,26].

### 3.4. Characteristics of CHC

#### 3.4.1. FTIR Spectral

The FT-IR spectra of CHC (Figure 2) showed that the characteristic frequency region of its FT-IR spectrum was from 3500 to 1350 cm^−1^, which was due to the many distinctive vibrational modes or group frequencies of the amide groups within the protein molecule [27]. The FTIR spectrum is similar to type I collagen, consistent with the above identification results. More specifically, all major bands associated with collagen-related functional groups were detected in CHC as the amide A band (3409–3280 cm^−1^), amide B band (2972 cm^−1^), amide I band (1658 cm^−1^), amide II band (1554 cm^−1^), and amide III band (1235 cm^−1^). The amide I and amide II bands are the leading ones identifying collagen. The amide I band is directly related to the collagen backbone conformation, which reflects the change in ν_C=O_ stretching vibration on the peptide backbone [28]. The amide I band of CHC had a higher wave number than the amide I band of marine fish collagen. This indicated that the peptide chain backbone of camel-hoof-source collagen was better ordered, and the triple helix structure was stable [29]. The amide II and III bands are characteristic frequency regions that occur after coupling N-H bending vibrations (δ_NH_) and C-N stretching vibrations (ν_C━N_). They are also distinct absorption regions for the secondary structure of proteins [26]. The presence of the amide II band suggests that the secondary collagen structure in CHC is stable. There is a minimal amount of irregular curling transition of CHC at low frequencies, which is the cause of the amide II band peak’s lower intensity [30].

N-H stretching bands (the amide A band of aggregates) and asymmetric stretching vibrations (ν^as^ _C-H_, amide B band) were also observed. The wider peak shape of the amide A band is due to the red shift of the CHC molecule’s vibrational frequency due to the N-H bond’s multi-polymerization with hydrogen bonding [31,32]. Since the amide B band’s telescopic vibration frequency exhibits a spectrum blue shift and its absorption intensity is weaker than that of the amide A band, the CHC may include more hybridized molecules, which would raise the bond energy and reduce the bond length of the amide B band [28]. The sample also has a peaked shape in the 1340 cm^−1^ band, representing the CHC showing the characteristics of proline side-chain rocking vibration [2]. In addition, a C-N stretching vibration at 1409 cm^−1^ suggests an interaction between the amide group and the molecule of the triple helix structure [33]. No absorption band corresponding to the carboxyl functional group was found in CHC (1740–1720 cm^−1^), consistent with the lyophilized pure collagen results previously reported in the literature [19]. Thus, the above analyses suggest that CHC retains a relatively intact triple-helical structure and has a high degree of purity.

#### 3.4.2. Circular Dichroism (CD) Spectra

Circular dichroism spectroscopy is one of the most important methods for studying protein conformation and determining the secondary structure of collagen. Figure 3 demonstrates the CHC CD spectrum. Natural collagen has a triple helix structure and a CD with a positive absorption peak near 225 nm and a negative absorption peak near 197 nm [34]. CHC shows strong positive and negative absorption at 221 nm and 204 nm, respectively, consistent with the triple helix conformation characteristic of the protein. If the collagen is completely denatured, the positive absorption peak at 220 nm disappears entirely, and the negative absorption peak is red-shifted [31]. This suggests that CHC has a complete collagen triple helix structure.

### 3.5. Emulsification and Emulsion Stability

Collagen emulsion stability and emulsification are influenced by pI, mass concentration, and ambient ionic concentration [35]. Figure 4a shows that the emulsification increased with increasing concentration of CHC, while the emulsification stability gradually decreased. The maximum emulsification was attained at a concentration of 0.5%, after which there was a small decline. The adsorption force between collagen molecules gradually increases as the concentration rises. A tight interfacial membrane layer will form when it reaches a certain critical value. Even if the concentration continues to grow, the emulsification ability no longer shows apparent changes [36]. Figure 4a shows that the emulsification stability of CHC continues to decrease with increasing concentration. This indicates that collagen surface tension increased due to macro-molecular aggregation and decreased hydrophobicity [33].

Figure 4b shows that the trend of emulsification and emulsion stability of CHC was similar for increasing environmental pH. Both emulsification and emulsion stability reached high values at pH 3–4 and gradually decreased with the lowest emulsification (50.10 ± 1.03%) and emulsion stability (58.61 ± 1.20%) at pH 7. It increased slightly at pH 7–10. This indicates that the isoelectric point of CHC is near pH 7, which reduces the exposed hydrophilic groups in the molecule, decreasing the ability of the substance to combine with water and falling solubility [34]. The collagen molecules are well dispersed in the pH range away from the isoelectric point, allowing them to move faster toward the interfacial membrane and increase emulsification [9]. Higher pH levels cause camelid collagen molecules’ net negative charge to increase, which causes the chains to repel one another more strongly. This phenomenon results in a reduction in the process of emulsification and the stability of emulsions [37].

Figure 4c shows that the emulsification of CHC showed an increasing trend within 0.1–0.5% NaCl concentration, with high values of both emulsification and emulsion stability of CHC at 0.5% NaCl concentration. This may be because an increase in NaCl concentration weakens the attraction that collagen molecules have for one another, allowing salt solubility to occur and for oil droplets to adsorb on the collagen interface [5]. To increase the stability and emulsification of collagen, salt ions compress the thickness of the diffused bilayer as the concentration rises. This decreases the potential on the surface of the emulsion droplets, which lowers the repellent force barrier of the emulsion system and makes it easier for the droplets to aggregate, which reduces the stability and emulsification of collagen [3,38].

### 3.6. Moisture Absorption and Moisturizing Properties

Figure 5 displays the humectancy and hygroscopicity curves of CHC and glycerol. The change in the humectancy of CHC was slightly more significant than that of glycerol but lower than that of fish scale collagen and camel skin collagen [16,39]. The humectancy of CHC and glycerol steadily decreased with time (Figure 5a). After 1h of placement, the moisture retention rate of CHC and glycerol significantly reduced to 75.67 ± 1.54% and 80.37 ± 1.66%, respectively. After 24 h of placement, the difference in humectancy between CHC (52.06 ± 1.55%) and glycerol (59.45 ± 2.21%) was small and continued for a more extended time, up to 48 h. The change in their humectancy was not significant. This is because collagen contains an abundance of hydrophilic amino acids, like glycine, hydroxyproline, and hydroxylysine, that prevent water loss [40].

At 30 °C and 80% humidity, the hygroscopicity of CHC and glycerol increased over time (Figure 5b). According to earlier research, hygroscopicity is influenced by the sample’s molecular makeup, the type or quantity of exposed hydrophilic groups, and the extraction technique [41]. The increase in the hygroscopicity of CHC was significantly weaker than that of glycerol after 4 h of placement. At 48 h of placement, it reached 8.57 ± 0.13%, higher than snakehead skin collagen [32]. This indicates that the number and type of hydrophilic groups exposed by CHC molecules are higher than those of soft-shelled turtle calipash and camel skin [13,16].

### 3.7. Rheological Analysis

The steady-state viscosity of the CHC solution versus the shear rate is shown in Figure 6. The steady-state density of the CHC solution decreased with increasing shear rate but increased with increasing solution concentration. The more distributed chain particles roll and spin to compact into a group when the shear rate increases due to shear stress between the flow layers, lowering the number of physical cross-linking points and decreasing viscosity [5,42]. A total of 0.1–0.9% of the CHC solution showed typical mimetic plasticity (Figure 6a). The collagen molecules were entangled at low shear rates and formed more physical cross-linking sites, increasing the solution’s apparent viscosity [42]. Previous studies have reported that the higher the relative molecular mass of straight-chain polymer molecules, the higher the plasticity [16].

Thus, the CHC solution exhibits a shear-thinning rheological behavior and is a pseudoplastic fluid. The viscous modulus (G″) and elastic modulus (G′) of CHC solutions increased with increasing shear frequency (Figure 6b). In the 0.01–68.13 Hz range, the G′ of CHC solution increased more than that of G″. In the 0.01–14.68 Hz range, the G″ of the CHC solution was more significant than G′. The solution was viscous-dominated [2]; with the increase in shear frequency, the CHC solution entered into a smooth region (G″ < G′). The intersection of G″ and G′ gradually increased with increasing concentration, and the corresponding frequencies were mainly distributed in 14.68–21.54, 14.68–21.54 Hz, 31.62–46.42 Hz, 46.42–68.13 Hz, and 46.42–68.13 Hz. The findings were superior to earlier studies on bovine collagen [21], which could be attributed to the camel’s colossal body weight, hostile environment, camel hoof protein composition, and molecular structure [11]. Figure 6c shows the changes in tan δ as a function of frequency. The loss modulus tanδ is the critical value for the transition of the sample from solid to liquid behavior; the smaller the value of tanδ, the more pronounced the elastic behavior of the sample [16]. The tan δ of the CHC solution tested at 0.01–100 Hz is not much different, and most of them are in the range of 0–1. This implies that flexibility is essential in the CHC solution system. As a result, the 0.1–0.9% CHC solution might have a stable liquid network structure.

## 4. Conclusions

In conclusion, the current study found that camel hoof tissue is a natural terrestrial source of collagen extraction. CHC proteomics revealed 13 collagen molecular components and one collagen enhancer. The results of this study have not been previously reported. The CHC samples had the most ANK and type I collagen, followed by type II and VI collagen. They had the spectroscopic and amino acid characteristics of typical collagen. In addition, CHC showed good emulsification stability, humectancy, hygroscopicity, and stable rheological properties. More research is needed to determine camel-derived collagen in vivo and in vitro metabolism. CHC exhibits substantial potential as a biomaterial, offering a range of possible applications, including developing collagen-peptide-containing products.

## Figures and Tables

**Figure 1 foods-12-03303-f001:**
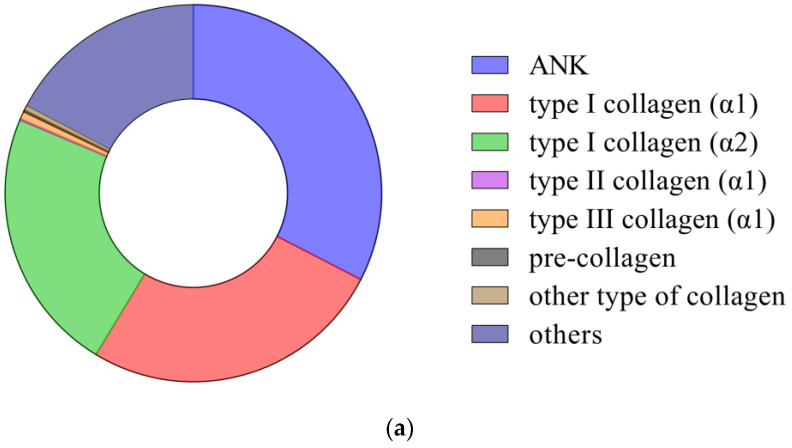
Protein composition, molecular weight, and pI distribution in CHC and identified collagens. (**a**) Protein composition; (**b**) molecular weight; (**c**) pI. Other types of collagen are represented: type IV collagen (α2), type V collagen (α1, α2, α3), type VI collagen (α1, α2, α3), and type Ⅺ collagen (α1, α2).

**Figure 2 foods-12-03303-f002:**
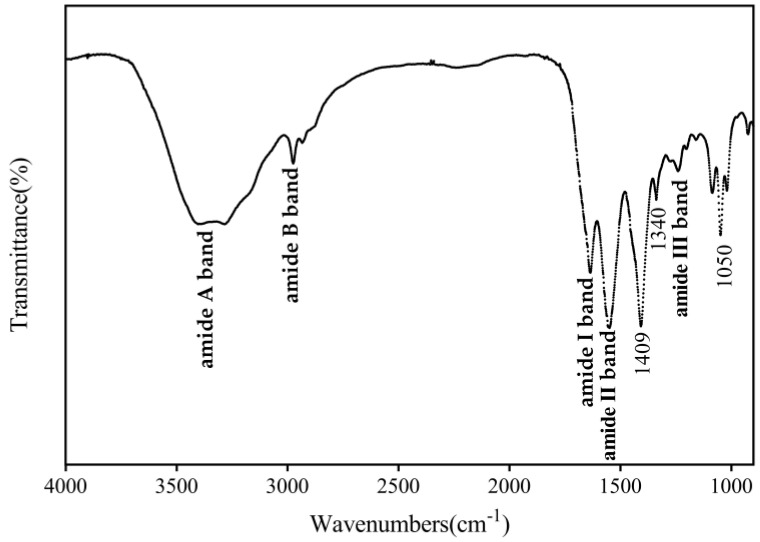
FTIR spectrum of CHC.

**Figure 3 foods-12-03303-f003:**
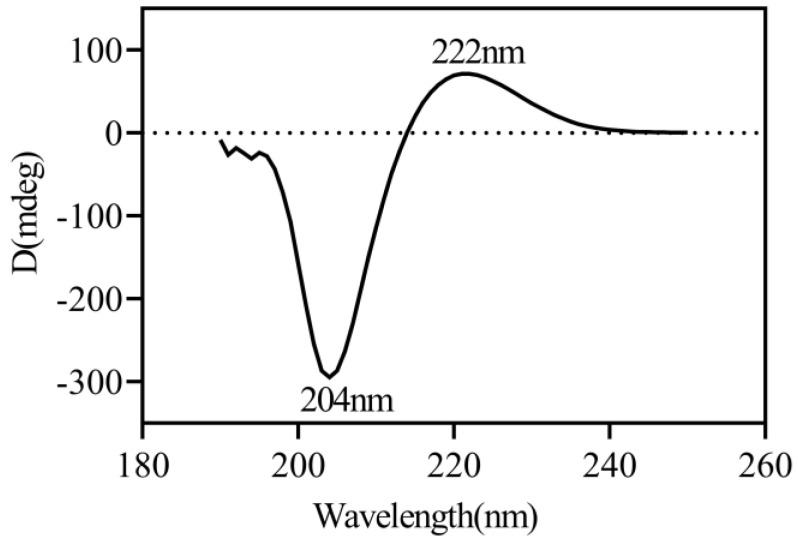
Circular dichroism spectra of CHC.

**Figure 4 foods-12-03303-f004:**
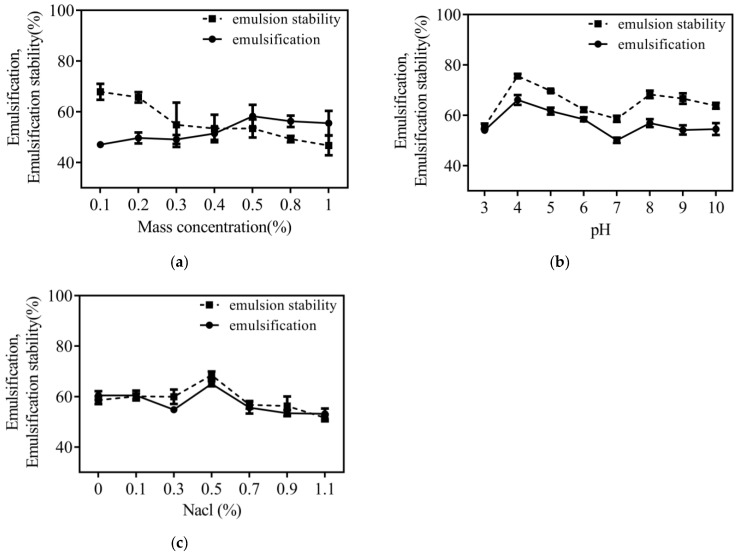
Mass concentration, pI, and ionic concentration on emulsification and emulsion stability. (**a**) Mass concentration; (**b**) pI; (**c**) ionic concentration.

**Figure 5 foods-12-03303-f005:**
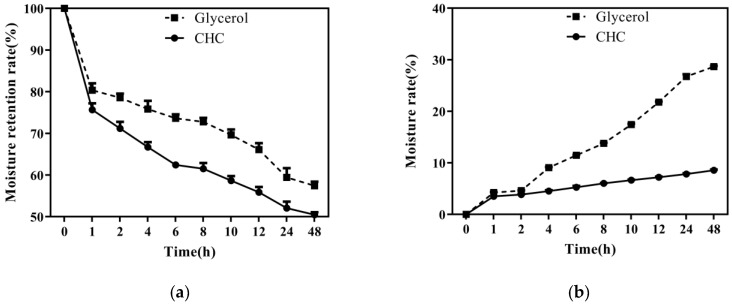
Humectancy and hygroscopicity curves of CHC. (**a**) Moisture absorption; (**b**) moisture retention.

**Figure 6 foods-12-03303-f006:**
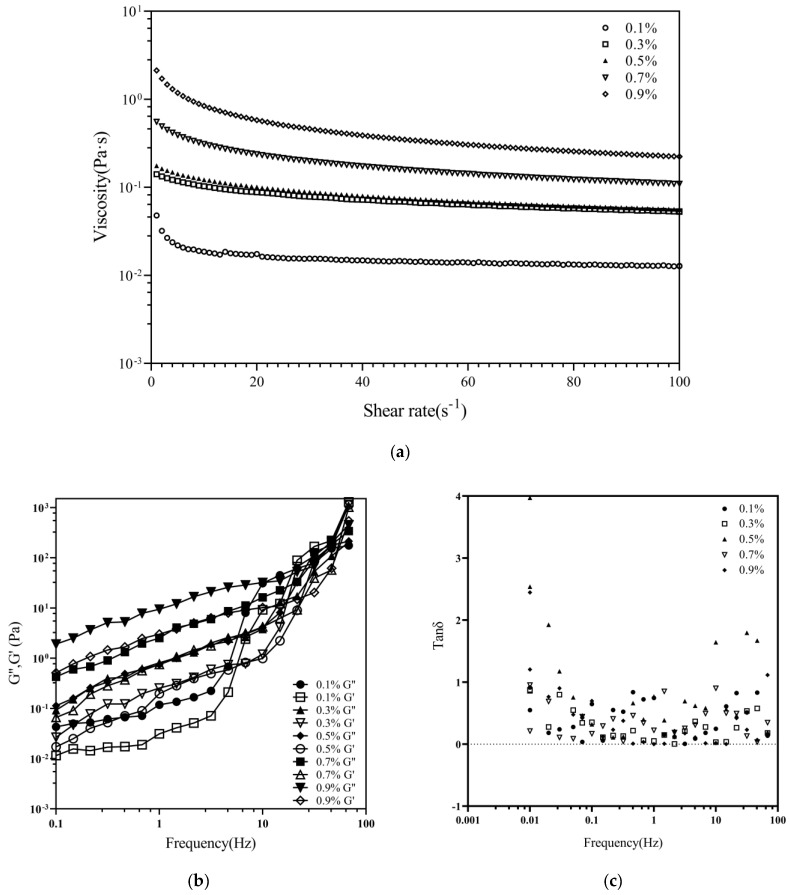
Steady-state viscosity, modulus of viscous elasticity, and tan δ of CHC solution with different concentrations. (**a**) Steady-state viscosity; (**b**) modulus of viscous elasticity; (**c**) tan δ.

**Table 1 foods-12-03303-t001:** Top 10 proteins and all collagen types identified in CHC samples.

Protein Description	Gene Names	Molecular Mass (kDa)	Sequence Coverage (%)	Sequence Length	Unique Peptides	iBAQ ^1^
*Ankyrin repeat and SOCS box containing 9 (ANK)	ASB9	31.447	4.5	287	1	2.74 × 10^9^
*Collagen alpha-1(I) chain	COL1A1	138.94	27.3	1463	29	2.19 × 10^9^
*Collagen alpha-2(I) chain	COL1A2	120.28	17.7	1274	22	1.91 × 10^9^
*Pseudouridylate synthase 7 homolog	PUS7	74.478	2.4	658	1	4.33 × 10^8^
*Multidrug resistance-associated protein 1	ABCC1	163.67	0.5	1464	1	1.41 × 10^8^
*G1 to S phase transition 1	GSPT1	66.015	1.7	605	1	1.16 × 10^8^
*Complement C1q like 1	C1QL1	26.468	2.7	258	1	1.10 × 10^8^
*Diacylglycerol kinase	DGKZ	126.46	0.8	1145	1	1.04 × 10^8^
*Actin, aortic smooth muscle	ACTA2	41.776	34.9	375	11	7.61 × 10^7^
*Collagen alpha-1(III) chain	COL3A1	138.44	5.6	1466	7	4.42 × 10^7^
Procollagen C-endopeptidase enhancer	PCOLCE	48.21	1.3	451	1	1.35 × 10^7^
Collagen alpha-1(II) chain	COL2A1	141.83	2.2	1487	3	9.56 × 10^6^
Collagen type VI alpha 3 chain	COL6A3	339.58	2	3132	7	7.43 × 10^6^
Collagen type VI alpha 1 chain	COL6A1	108.67	2.2	1027	2	6.05 × 10^6^
Collagen alpha-1(XI) chain (Fragment)	COL11A1	89.147	1.8	911	2	4.96 × 10^6^
Collagen type VI alpha 2 chain	COL6A2	97.135	3.2	917	3	4.91 × 10^6^
Collagen type V alpha 2 chain	COL5A2	123.7	2.2	1240	3	4.48 × 10^6^
Collagen type V alpha 1 (Fragment)	COL5A1	93.742	2	1014	2	3.24 × 10^6^
Collagen alpha-2(XI) chain	COL11A2	172.24	1.4	1736	3	2.08 × 10^6^
Collagen alpha-2(IV) chain	COL4A2	147.58	0.5	1496	1	3.35 × 10^5^
Collagen type V alpha 3 chain	COL5A3	172.07	1	1745	2	3.72 × 10^5^
Total identified proteins					272	8.40 × 10^9^

^1^ iBAQ (intensity-based absolute protein quantification). * Top 10 proteins in CHC samples.

**Table 2 foods-12-03303-t002:** Amino acid composition and contents in CHC (g/100 g sample).

Amino Acids	Content	Amino Acids	Content
Threonine (Thr)	1.63 ± 0.09	Glutamic acid (Glu)	7.28 ± 0.46
Isoleucine (Ile)	1.02 ± 0.05	Aspartic acid (Asp)	4.14 ± 0.20
Leucine (Leu)	2.18 ± 0.14	Serine (Ser)	2.53 ± 0.13
Phenylalanine (Phe)	1.49 ± 0.09	Tyrosine (Tyr)	0.33 ± 0.02
Valine (Val)	1.84 ± 0.07	Glycine (Gly)	15.11 ± 1.78
Lysine (Lys)	2.28 ± 0.13	Alanine (Ala)	6.20 ± 0.38
Methionine (Met)	0.31 ± 0.03	Arginine (Arg)	5.83 ± 0.36
Histidine (His)	0.49 ± 0.02	Proline (Pro)	9.54 ± 1.06
Essential amino acid (EAA)	11.58 ± 1.95	Hydroxyproline (HYP)	6.06 ± 1.16
Total amino acids	62.21 ± 3.10		

## Data Availability

All data are available within the article.

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
