# Peer review of "Proteomic Identification and Characterization of Collagen from Bactrian Camel (Camelus bactrianus) Hoof"

_foods, 2023, doi:10.3390/foods12173303_

Round 1
Reviewer 1 Report
Comments and Suggestions for Authors
This paper focuses on the proteomic identification and characterization of collagen from Bactrian camel hoof extracts (CHC). The authors utilized LC-MS/MS technique to explore the structural characterization and main functional properties of CHC, including rheological properties, emulsification ability and stability, humectancy, and hygroscopicity. The paper does not mention any comparisons with collagen from other types of camels.
Specific comments:
1. The abstract is not written in a comment way. Abstract usually starts with a bit of background, followed obejctives, methods and conclusion.
2. Provide the paraemeters used in amino acid analysis
3. What si UA?
4. Eplain why HCD is the method of choice and not the common method CID.
5. Where did you get the camel database? is there any update on the entries since 2020?
6. table1: what do you mean by "majority IDs"
7. Discussion on hydroxyproline content is lacking.
8. Didnt you need animal ethics approval to conduct this project? Please provide proof.
9. Please supply the sequences of detected collagens as appendix.
Comments on the Quality of English Language
Certain parts are hard to understand. The manuscript needs to be reviewed by an editor or a native speaker.
Reviewer 2 Report
Comments and Suggestions for Authors
The authors characterize a collagen preparation derived from camel hoof. They conclude that “camel hoof tissue is a natural terrestrial source of collagen extraction”, and that “CHC is a promising source of biomaterials, with uses ranging from developing collagen peptide products.” There are some serious flaws in the methods and interpretations that must be addressed. These are outlined below, and the writing should be improved, some grammatical errors are also pointed out below.
The authors should consider adding a purified/commercial source of collagen as a control for each of the assays they utilize in this work, as they mention on lines 410-411 “They had the spectroscopic and amino acid characteristics of higher-purity collagen.” Then at each step they would have a ‘known’ that they could compare to the values they observe for their CHC samples.
Lines 9-12 The first sentence in the abstract is too long and is not clear. LC-MS/MS technique was used for the proteomic identification of Bactrian camel (Camelus bactrianus) hoof collagen, but it was not used to explore the other properties (rheological properties, emulsification ability and stability, humectancy and hygroscopicity) mentioned in the first sentence.
The CHC described in section 2.2 is a result of pepsin treated hoof material and centrifugation, so was the mass spectrometry data analyzed using digestion of both pepsin and trypsin cleavage sites? This should be noted in lines 131-134 of the M&M and lines 186-195 of the results.
Also what is the meaning of the sentence on line 91-92 “The supernatant was centrifuged at 10,000 g for 20 min to separate the residues.” ?
Also, does the pepsin remain in the CHC preparation?
The authors should include SDS-PAGE analysis of the CHC. This would provide visual evidence of the size and uniformity of the proteins in the CHC preparation.
On lines 181-184 rearrange the sentences as follows, “The appropriate concentration of glacial acetic acid solution facilitates the extraction of CHC. This may be because the low acidic environment interfered with the stability of the salt bonds and Schiff base structure between collagen molecules, facilitating collagen solubilization [16]. In this experiment, the collagen yield of CHC was 35.77 ± 2.06%. This is similar to the results of collagen extraction from Bactrian camel skin [15].”
The sentence on lines 186-188 is worded awkwardly, consider revising.
Importantly, in Table 1, 8 of the proteins listed had only one peptide ‘hit’, and this is not acceptable. “A single peptide mass measurement is typically not matched uniquely with a single protein species and is therefore not sufficient to identify a protein (the probability for more than one protein identified =1).” see David Fenyö, Jan Eriksson, Ronald Beavis. Mass spectrometric protein identification using the global proteome machine. Methods Mol Biol. 2010;673:189-202. doi: 10.1007/978-1-60761-842-3_11. PMID: 20835799 PMCID: PMC3757509 DOI: 10.1007/978-1-60761-842-3_11 for more information and as a reference for your manuscript. To have high confidence in a protein identification using mass spectrometry at least two unique peptides from the same protein should be identified.
Table 1, further, are the peptides listed unique ‘hits’ (significant peptides that are unique to each collagen isoform). That is, are any of the collagen peptides shared among the different collagen isoforms? This table and the results text need to be revised and rewritten, or the authors should repeat the mass spectrometry to increase the number and coverage of any protein/peptide data they are presenting.
Figure 1 (if I’m interpreting correctly) is also based upon the Table 1/mass-spectrometry data, and so this figure should be revised accordingly based upon my comments about Table 1 and the MS data. Figure 1 should be based upon a Coomassie and/or silver stained SDS-PAGE gel.
If I understand correctly, according to the way the M&M were written, the FTIS and CD were done using the pepsin treated hoof material described in section 2.2. If so, it makes the FTIS and CD data difficult to interpret since the protein is pepsin treated and it’s not clear what utility this data provides. The authors need to clarify if the proteins are full length or pepsin derived peptides.
Figure 2 should be revised. There is no figure legend so the authors should consider incorporating the text on lines 274-276 “amide A band (3409-3280 cm-1), amide B band (2972 cm-1), amide I band (1658 cm-1), amide II band (1554 cm-1) and amide III band (1340 cm-1)” as part of a figure legend and labeling the bands directly on the figure plot. Also, is amide B band 2971 or 2912 cm-1?
It’s not clear what you are measuring in Figure 3? CD is most often used for purified proteins, and the authors indicate that the CHC is composed of numerous proteins. I don’t believe this figure has any clear interpretation. The authors indicate on lines 311-314 that “If the collagen is completely denatured, the positive absorption peak at 220 nm disappears entirely, and the negative absorption peak is red-shifted [31]”. They should test this directly on their CHC prep using the methods described in reference 31.
Line 318-319 if the authors are only referencing published material this sentence should be changed to the following “Collagen emulsion stability and emulsification are influenced by pI, mass concentration, and ambient ionic concentration [35].”
Line 320, 329, 332, 337, 343, 344 “Emulsification” should not be capitalized
What pH and salt concentration was used for Figure 4a? Similar what salt and mass were used for 4B, and what mass and pH for 4C?
Why is glycerol included in Figure 5? Why not compare with a purified/commercial source of collagen?
Line 414-415 Revise this sentence to the following, “CHC is a promising source of biomaterials, with potential uses including the development of collagen peptide containing products.”
Comments on the Quality of English Language
see my authors comments
Round 2
Reviewer 2 Report
Comments and Suggestions for Authors
The authors have adequately addressed each of the points raised in the review.
Comments on the Quality of English LanguageNA
